# Difference in predictors and barriers to arts and cultural engagement with age in the United States: A cross-sectional analysis using the Health and Retirement Study

**Meg Fluharty**[1]*, **Elise Paul**[1], **Jessica Bone**[1], **Feifei Bu**[1], **Jill Sonke**[2], **Daisy Fancourt**[1]

**1** Research Department of Behavioural Science and Health, Institute of Epidemiology & Health, University College London, London, United Kingdom, **2** Center for Arts in Medicine, University of Florida, Gainesville, Florida, United States of America

\* meg.fluharty@gmail.com

## Abstract

### Introduction

Arts and cultural engagement are associated with a range of mental and physical health benefits, including promoting heathy aging and lower incidence of age-related disabilities such as slower cognitive decline and slower progression of frailty. This suggests arts engagement constitutes health-promoting behaviour in older age. However, there are no large-scale studies examining how the predictors of arts engagement vary with age.

### Methods

Data from the Health and Retirement Study (2014) were used to identify sociodemographic, life satisfaction, social, and arts appreciation predictors of (1) frequency of arts engagement, (2) cultural attendance, (3) difficulty participating in the arts, and (4) being an interested non-attendee of cultural events. Logistic regression models were stratified by age groups [50–59, 60–69, ≥70] for the frequency of arts participation outcome and [50–69 vs ≥70] all other outcomes.

### Results

Findings indicated a number of age-related predictors of frequent arts engagement, including gender, educational attainment, wealth, dissatisfaction with aging, and instrumental activities of daily living (iADL). For cultural event attendance, lower interest in the arts predicted lack of engagement across age groups, whereas higher educational attainment and more frequent religious service attendance became predictors in older age groups (≥ 70). Adults in both age groups were less likely to report difficulties engaging in the arts if they had lower neighbourhood safety, whilst poor self-rated health and low arts appreciation also predicted reduced likelihood of this outcome, but only in the younger (50–69) age group. Adults in the older (≥ 70) age group were more likely to be interested non-attendees of cultural events if they had higher educational attainment and less likely if they lived in neighbourhoods with low levels of safety.

**Data Availability Statement:** Data is publicly available on the Health and Retirement Study website (https://hrs.isr.umich.edu/).

**Funding:** The EpiArts Lab, a National Endowment for the Arts Research Lab at the University of Florida, is supported in part by an award from the National Endowment for the Arts (https://www.arts.gov/) [Award: 1862896-38-C-20]. The opinions expressed are those of the authors and do not represent the views of the National Endowment for the Arts Office of Research & Analysis or the National Endowment for the Arts. The National Endowment for the Arts does not guarantee the accuracy or completeness of the information included in this material and is not responsible for any consequences of its use. The EpiArts Lab is also supported by the University of Florida, the Pabst Steinmetz Foundation, and Bloomberg Philanthropies. DF is supported by the Wellcome Trust (https://wellcome.org/)[205407/Z/16/Z]. The funders had no role in study design, data collection and analysis, decision to publish, or preparation of the manuscript.

**Competing interests:** The authors have declared that no competing interests exist.

## Conclusions

Our results suggest that certain factors become stronger predictors of arts and cultural engagement and barriers to engagement as people age. Further, there appear to be socio-economic inequalities in engagement that may increase in older ages, with arts activities overall more accessible as individuals age compared to cultural engagement due to additional financial barriers and transportation barriers. Ensuring that these activities are accessible to people of all ages will allow older adults to benefit from the range of health outcomes gained from arts and cultural engagement.

## Introduction

A recent report by the World Health Organization found evidence that engaging in the arts has a range of health benefits, from supporting social determinants of health and playing a role in the prevention of mental and physical illness, to assisting in the management and treatment of various health conditions [1]. Arts engagement is a ubiquitous human behaviour that involves a number of 'active ingredients' that are known to be health promoting, including sensory activation, cognitive stimulation, a reduction in sedentary behaviours, and social interaction [1, 2]. These ingredients have been found to trigger a wide range of psychological, physiological, social and behavioural mechanisms that can, in turn, affect mental and physical health and health behaviours [3]. Specifically, there has been increasing research in recent years on the benefits of arts engagement in older age, with studies showing associations with reduced loneliness [4], lower incidence of depression [5], higher subjective wellbeing [6], slower cognitive decline [7], a lower risk of developing dementia [8], lower incidence of chronic pain [9], lower incidence of and slower progression of frailty [10], lower risk of developing age-related disability [11], lower incidence of chronic diseases such as cardiovascular disease and cancers [12], and lower mortality rates [13]. As such, arts engagement can be seen to constitute a health-promoting behaviour in older age.

However, evidence from previous studies has suggested some age related barriers to arts participation (such as poor health, lack of time, and interest) and attendance of arts events (such as costs, no one to attend with, and lack of transportation) [14]. In the United Kingdom (UK) studies have shown noticeable declines in arts participation beginning around age 65 [14], with the steepest declines for individuals over 85 [14, 15]. Similar trends are seen with cultural attendance (such as visiting museums, historic parks and gardens) alongside declines in physical activity [16]. There have been reports of declining arts participation with age in the United States (US) as well [17–19]. Gender differences in engagement have also been observed, which may increase with age, with women in older age groups more likely to attend arts events than men [16]. A number of sociodemographic disparities in arts and cultural engagement have also been identified, which are often amplified in older age [14, 20]. These include illness and disability [21], living alone [21], lower educational attainment [21–23], ethnic minority status [24], and living in areas with high levels of poverty [16]. Therefore, it is important to understand how these previously identified sociodemographic disparities in access to the arts may change as people age.

Additionally, it has been demonstrated that there is little evidence of predictors of arts and cultural engagement changing *over time*, with predictors in the 1990s showing very high levels of similarity to predictors in the 2010s [25]. However, it is currently unclear whether predictors

of arts engagement vary *as people age;* in other words, whether certain demographic, socioeconomic, or health-related factors cause more barriers to engagement in later life compared to in middle age. Further, there has been little distinction in the past between the predictors of arts engagement vs predictors of *wanting* to engage in the arts but not being able to. This is a crucial distinction as it differentiates between a lack of interest or motivation vs a lack of capability or opportunity [26]. A social gradient associated with interested non-attendance in the arts has been observed, with a number of additional barriers to engagement such as finances, lack of time, and transportation difficulties [25]. But this research has focused exclusively on attending arts events (such as classical music or opera performances) rather than other types of arts engagement (such as participatory art, i.e. production of art versus attendance and appreciation) and looked across adulthood rather than at older age specifically [25]. Therefore, the current study used a large nationally representative cohort study of adults over fifty years of age in the US (the Health and Retirement Study [27]) to explore in much greater depth (i) how predictors of arts engagement differ across different age groups and (ii) the consistency between predictors of engagement amongst all participants compared to participants who specifically wanted to engage but reported facing barriers. We hypothesised that sociodemographic, life satisfaction, social, health, and interest in the arts related factors would aid in predicting (1) arts and cultural engagement as people across different age groups, and (2) barriers to arts and cultural engagement across different age groups. Overall, we hypothesised that better socioeconomic circumstances, higher life satisfaction, better social and health functioning, and increased interest in the arts would be associated with more arts engagement as people aged, while the opposite would be associated with increased barriers to the arts as people aged.

## Methods

### Participants

Participants were drawn from the Health and Retirement Study (HRS), a nationally representative study in the United States of over 37,000 individuals over the age of 50 [27]. The study was initiated by the National Institute on Aging (NIA) and conducted by the Institute for Social Research (ISR) at the University of Michigan to track the Baby Boom generation's transition from work to retirement. The HRS covers a range of topics including income and wealth, health, cognition, use of healthcare services, work and retirement, and family [27]. The initial HRS cohort was interviewed for the first time in 1992 and followed-up every two years with data collection still ongoing. Over the years, five other studies were merged with the initial HRS cohort; Asset and Health Dynamics among the Oldest Old (AHEAD), the Children of the Depression (CODA), the War Babies, Early Baby Boomers (EBB), and Mid Baby Boomers (MBB) [27]. HRS replenishes the sample every six years with younger cohorts to ensure the data continues to provide a fully representative sample of individuals over the age of 50 in the United States; more details on study design are described elsewhere [27].

### Sample

Included participants were required to have responded to the enhanced interview questions on arts activities in HRS at wave 12 and/or participated in the wave 12 Culture and Arts module. Two samples were derived from the total merged HRS cohort for the present study. First, of the total number of participants recruited by the start of wave 12 in the year 2014 (N = 35,364), there were 21,525 known or presumed alive (via correspondence or family notification). Of these, there were 18,747 participants (87.1% response rate) [28] who took part in the core survey where one of our four arts outcomes, frequency of arts engagement, was measured. Within each wave, a rotating random 50% subsample is invited for an enhanced

interview (N = 9,459 in 2014), and following the interview a questionnaire is left for participants to complete and mail back to the study offices [29]. A total of 7,541 completed the questionnaire in 2014, and there were 7,523 participants who completed arts activities questions who thus formed our sample for frequency of arts engagement analyses.

Second, a random subsample of 1,500 respondents in the core survey were invited to participate in the Culture and Arts module in this wave. Of the 1,500 respondents recruited, 1,496 participants (99.7%) took part. Of these, 1,465 (97.9%) had non-missing data on our other three outcomes. This became the sample for our analyses on difficulty participating in the arts, cultural attendance, and missed cultural events.

### Outcome variables

We focused on four outcome variables: frequency of arts participation, cultural attendance, difficulty participating in the arts, and missed cultural events. All four prompted participants to focus on the past 12 months. Frequency of arts participation was created by asking participants 'How often do you [do writing, bake or cook, sew or knit, read, do hobbies, participate in a community arts group]' with responses of 'several times a week,' 'once a week,' 'several times a month,' 'at least once a month,' 'not in the last month,' and 'never.' The most frequent response across questions was chosen, and answers were collapsed into 'weekly or more,' and 'less than weekly.' Cultural attendance in the past 12 months was assessed by asking participants 'In the past 12 months, did you go to a movie, art museum or gallery, crafts fair, or a live performance, such as a concert, play, or reading?' with responses of 'yes' and 'no.' Difficulty participating in the arts was operationalised from responses to the statement 'It is difficult for me to participate in the arts,' with responses of 'strongly agree,' 'agree,' 'neither agree or disagree,' 'disagree,' 'strongly disagree.' These were collapsed into 'neutral / disagree' and 'agree.' Our missed cultural events variable was created by asking participants 'In the last 12 months, was there an event of this type that you wanted to go to but did not,' with responses of 'yes' or no'. All outcomes were collapsed to binary due to small sample sizes in alternative categorisations.

### Predictor variables

We included eighteen predictor variables, framed as presented in the 2014 wave of the HRS survey instrument.

*Sociodemographic* variables were: (1) gender [men, women], (2) race/ethnicity [choices were offered as White (White/ Caucasian), Black/African American, Other ethnicity (American Indian or Alaskan Native, Asian or Pacific Islander, Other)], participations were required to choose one category and could not report multiple ethnicities, (3) marital status [married, unmarried (separated/divorced, widowed, and never married)], (4) educational attainment [none, high school/GED, college/postgraduate], (5) self-reported neighbourhood safety [excellent/good, fair/poor], (6) employment status [employed, unemployed/inactive, retired], and (7) wealth [quartiles]: total of all assets.

*Life satisfaction* variables were: (8) satisfaction with aging yes vs no] and (9) satisfaction with life [yes vs no].

*Social* variables were: (10) religious service attendance [none, monthly or less, weekly] and (11) frequency of seeing friends [yearly or less, monthly vs weekly]. Health factors included were: (12) depression [present vs not]: measured using an 8 item version of the Centre for Epidemiological Studies Depression Scale (CES-D) [30], with those scoring ≥4 categorised as having depression [31]. (13) smoking status [smoker, non-smoker]: created by collapsing ex-smokers into non-smokers, (14) self-rated health [good/excellent, fair/poor], (15) difficulties

with 'instrumental activities of daily living' (iADLs) [none, difficulties with activities, unable to do activities]: created by collapsing responses to activities including using maps, preparing hot meals, shopping for groceries, making phone calls, taking medicines, and paying bills/ tracking expenses [32]. (16) long term conditions [yes, none]: by indicating the presence of a number of disorders including complications from stroke, diabetes, lung disease, cancer, heart conditions, or other medical conditions. (17) cognition score [quartiles]: word recall summary score [28].

*Appreciation for the arts* was measured with: (18) interest in the arts index score (Cronbach's alpha = 0.95) which is a mean of seven statements on attitudes towards the arts including 'the arts are important to me', 'I enjoy the arts', 'I don't have any interest in the arts', 'the arts help me stay active and engaged', 'the arts help me socialise with family and friends', 'I like to take lessons or classes in the arts', and 'I feel a sense of appreciation for the arts'. Responses were rated on a scale from 1 (strongly agree) to 5 (strongly disagree). Higher scores reflect lower appreciation for the arts.

## Statistical analysis

First, logistic regression models were used to examine associations between sociodemographic, life satisfaction, social, health, and appreciation for the arts factors with either arts and cultural *engagement* or *barriers to engagement* for each of our four outcomes: (1) frequency of arts participation, (2) cultural event attendance, (3) difficulties participating in the arts, and (4) interested non-attendees. For our frequency of arts participation outcome, we also included age as a predictor variable for the first analysis (as we had more than two age groups. Next, all models were rerun stratified by age group. We stratified by three age groups [50–59, 60–69, ≥70] for the frequency of arts participation outcome and then stratified by two age groups [50–69 vs ≥70] for cultural event attendance, difficulty participating in the arts, and interested non-attendee status. All analyses were weighted according to age, race/ethnicity, education, and state in the US population to account for unequal sampling [33]. Multiple imputation by chained equations (MICE) was conducted to address missing data in exposures, resulting in 50 imputed datasets. We included participants without missing information on outcome variables in the imputation model (see S1 Table for pattern of missingness in study variables). All analyses were preformed using Stata 16 [34].

## Results

### RQ1: Predictors of arts participation and cultural attendance

**Arts participation.**   In the sample (N = 7,523) used to examine frequency of arts participation, 59.7% were women, 32.6% were educated to the college/postgraduate degree level, and 49.0% were retired. A majority (88.2%) reported weekly or more participation in the arts, and 11.8% reported less than weekly participation. Predictors for the total sample are shown in Table 1. In comparison to ages 50–59, the 70+ age group was most likely to participate at least weekly in arts (OR = 1.81; 95% CI = 1.24, 2.64) (Table 1).

When stratified by age, individuals aged 50–59 in the second lowest wealth quartile had a higher likelihood of weekly or more arts participation (Quartile 2: OR = 2.42; 95% CI = 1.32, 4.47) than those in the lower wealth quartile (Table 1). In adults aged 60–69, higher wealth (Q3: OR = 1.95; 95% CI = 1.05, 3.60) and seeing friends on a weekly basis (OR = 1.81; 95% CI = 1.05, 3.12) were associated with greater odds of participating weekly or more in the arts. Adults aged 70 and older in the highest cognition score quartile were more likely than those in the lowest quartile to participate in arts at least weekly (OR = 2.88; 95% CI = 1.24, 6.66). Those in both the 60–69 and 70+ age groups were more likely to participate at least weekly if they had

**Table 1. Age related differences in predictors of frequency of arts participation in the past 12 months from logistic regression models.**

| | Frequency of arts participation | | | | | | | | | | | |
|---|---|---|---|---|---|---|---|---|---|---|---|---|
| | Total sample [A] | | | Ages ≤ 59 [B] | | | Ages 60–69 [C] | | | Ages ≥ 70 [D] | | |
| | N = 7523 | | | N = 1827 | | | N = 2316 | | | N = 3380 | | |
| | OR | 95%CI | P | OR | 95%CI | P | OR | 95%CI | P | OR | 95%CI | P |
| **Age (Ref 50–59)** | - - | | | | | | | | | | | |
| 60–69 | 1.31 | 0.95 1.79 | 0.096 | | | | | | | | | |
| 70+ | **1.81** | **1.24 2.64** | **0.002** | | | | | | | | | |
| **Gender (Ref Female)** | - - | | | | | | | | | | | |
| Male | **0.51** | **0.40 0.66** | **<0.001** | 0.65 | 0.39 1.06 | 0.084 | **0.41** | **0.27 0.61** | **<0.001** | **0.44** | **0.31 0.64** | **<0.001** |
| **Race/Ethnicity (Ref White)** | - - | | | | | | | | | | | |
| Black/African American | 0.91 | 0.67 1.23 | 0.522 | 1.48 | 0.82 2.65 | 0.191 | 0.82 | 0.48 1.41 | 0.475 | 0.72 | 0.46 1.15 | 0.168 |
| Other ethnicity [including American Indian or Alaskan Native, Asian or Pacific Islander] | 0.90 | 0.60 1.34 | 0.602 | 1.50 | 0.67 3.36 | 0.323 | 0.77 | 0.41 1.44 | 0.415 | **0.51** | **0.28 0.92** | **0.026** |
| **Marital status (Ref Married)** | - - | | | | | | | | | | | |
| Unmarried | 1.01 | 0.78 1.31 | 0.934 | 1.14 | 0.69 1.89 | 0.619 | 0.96 | 0.61 1.50 | 0.849 | 0.92 | 0.63 1.35 | 0.673 |
| **Educational attainment (Ref None)** | - - | | | | | | | | | | | |
| High School/ GED | **1.74** | **1.29 2.35** | **<0.001** | 1.67 | 0.87 3.21 | 0.126 | 1.14 | 0.65 1.99 | 0.658 | **2.30** | **1.57 3.35** | **<0.001** |
| College / postgraduate | **3.02** | **1.97 4.64** | **<0.001** | 1.96 | 0.86 4.47 | 0.111 | **2.51** | **1.20 5.22** | **0.014** | **6.04** | **3.21 11.37** | **<0.001** |
| **Neighbourhood safety (Ref Good/excellent)** | - - | | | | | | | | | | | |
| Fair/Poor | 1.16 | 0.83 1.61 | 0.384 | 0.97 | 0.51 1.85 | 0.918 | 1.21 | 0.70 2.10 | 0.498 | 1.24 | 0.77 1.98 | 0.374 |
| **Employment status (Ref Employed)** | - - | | | | | | | | | | | |
| Unemployed/ Inactive | 0.94 | 0.67 1.32 | 0.728 | 0.95 | 0.56 1.61 | 0.858 | 0.89 | 0.52 1.53 | 0.681 | 0.51 | 0.23 1.11 | 0.088 |
| Retired | 1.01 | 0.74 1.37 | 0.960 | 1.51 | 0.52 4.42 | 0.453 | 1.00 | 0.65 1.54 | 0.991 | 0.59 | 0.29 1.20 | 0.145 |
| **Wealth, quartiled (Ref Quartile 1)** | - - | | | | | | | | | | | |
| Quartile 2 | **1.50** | **1.11 2.02** | **0.008** | **2.42** | **1.32 4.47** | **0.005** | 1.40 | 0.84 2.33 | 0.199 | 1.10 | 0.73 1.66 | 0.649 |
| Quartile 3 | **1.52** | **1.09 2.14** | **0.015** | 1.64 | 0.83 3.23 | 0.152 | **1.95** | **1.05 3.60** | **0.034** | 1.23 | 0.79 1.93 | 0.360 |
| Quartile 4 | **1.81** | **1.20 2.74** | **0.005** | 2.04 | 0.85 4.92 | 0.111 | 1.71 | 0.85 3.43 | 0.133 | 1.53 | 0.88 2.68 | 0.135 |
| **Satisfied with ageing (Ref Yes)** | - - | | | | | | | | | | | |
| No | **1.43** | **1.07 1.91** | **0.016** | 0.79 | 0.43 1.46 | 0.452 | **2.11** | **1.28 3.48** | **0.003** | **1.66** | **1.11 2.48** | **0.013** |
| **Satisfied with Life (Ref Yes)** | - - | | | | | | | | | | | |
| No | 0.99 | 0.76 1.29 | 0.948 | 0.97 | 0.58 1.61 | 0.905 | 0.82 | 0.51 1.32 | 0.418 | 1.18 | 0.80 1.73 | 0.400 |
| **See friends (Ref Yearly/less)** | - - | | | | | | | | | | | |
| Monthly | 1.11 | 0.80 1.54 | 0.550 | 1.11 | 0.56 2.21 | 0.768 | 1.03 | 0.61 1.75 | 0.907 | 1.17 | 0.74 1.84 | 0.507 |
| Weekly | **1.61** | **1.13 2.30** | **0.009** | 1.51 | 0.70 3.28 | 0.295 | **1.81** | **1.05 3.12** | **0.032** | 1.41 | 0.88 2.27 | 0.151 |
| **Attend religious services (Ref Yearly/Less)** | - - | | | | | | | | | | | |
| Monthly | 1.02 | 0.76 1.36 | 0.921 | 0.84 | 0.50 1.43 | 0.519 | 1.46 | 0.92 2.32 | 0.111 | 0.68 | 0.45 1.02 | 0.064 |
| Weekly | 1.27 | 0.94 1.72 | 0.115 | 0.96 | 0.52 1.77 | 0.886 | 1.26 | 0.76 2.09 | 0.369 | 1.47 | 0.97 2.21 | 0.070 |
| **Depression CES-D (Ref None)** | - - | | | | | | | | | | | |
| Present | 0.81 | 0.59 1.12 | 0.203 | 0.62 | 0.34 1.14 | 0.126 | 1.11 | 0.63 1.95 | 0.711 | 0.75 | 0.49 1.16 | 0.197 |
| **Smoker** | - - | | | | | | | | | | | |
| Yes | 1.04 | 0.69 1.56 | 0.842 | 0.76 | 0.41 1.41 | 0.383 | 1.04 | 0.57 1.89 | 0.899 | 1.56 | 0.84 2.89 | 0.159 |
| **Self-rated health (Ref Good/excellent)** | - - | | | | | | | | | | | |
| Fair/Poor | 1.09 | 0.83 1.43 | 0.547 | 1.15 | 0.64 2.05 | 0.649 | 1.18 | 0.72 1.93 | 0.510 | 0.93 | 0.65 1.32 | 0.680 |
| **iADL (Ref None)** | - - | | | | | | | | | | | |
| Difficulties with activities | **0.74** | **0.56 0.97** | **0.032** | 0.88 | 0.47 1.67 | 0.704 | 0.70 | 0.43 1.13 | 0.140 | **0.64** | **0.45 0.91** | **0.012** |
| Unable to do activities | **0.31** | **0.17 0.55** | **<0.001** | 0.34 | 0.08 1.49 | 0.154 | **0.32** | **0.11 0.95** | **0.039** | **0.31** | **0.14 0.67** | **0.003** |
| **Long term conditions (Ref None)** | - - | | | | | | | | | | | |
| Yes | 0.88 | 0.68 1.15 | 0.356 | 0.88 | 0.55 1.42 | 0.595 | 0.94 | 0.61 1.43 | 0.760 | 0.79 | 0.51 1.22 | 0.284 |

(*Continued*)

**Table 1.** (Continued)

| | Frequency of arts participation | | | | | | | | | | | |
| --- | --- | --- | --- | --- | --- | --- | --- | --- | --- | --- | --- | --- |
| | Total sample [A] | | | Ages ≤ 59 [B] | | | Ages 60–69 [C] | | | Ages ≥ 70 [D] | | |
| | N = 7523 | | | N = 1827 | | | N = 2316 | | | N = 3380 | | |
| | OR | 95%CI | P | OR | 95%CI | P | OR | 95%CI | P | OR | 95%CI | P |
| **Total cognition score, quartiled (Ref Quartile 1)** | - - | | | | | | | | | | | |
| Quartile 2 | 1.25 | 0.93 | 1.68 | 0.141 | 1.36 | 0.66 | 2.83 | 0.407 | 1.34 | 0.79 | 2.27 | 0.280 | 1.03 | 0.72 | 1.48 | 0.861 |
| Quartile 3 | 1.17 | 0.80 | 1.71 | 0.414 | 0.97 | 0.42 | 2.22 | 0.942 | 1.53 | 0.82 | 2.85 | 0.177 | 1.03 | 0.59 | 1.83 | 0.910 |
| Quartile 4 | **1.68** | **1.09** | **2.61** | **0.020** | 1.61 | 0.66 | 3.94 | 0.294 | 1.86 | 0.93 | 3.72 | 0.079 | **2.88** | **1.24** | **6.66** | **0.014** |
| **Arts index** | **0.63** | **0.44** | **0.89** | **0.009** | **0.61** | **0.37** | **0.99** | **0.047** | **0.61** | **0.38** | **1.00** | **0.048** | **0.65** | **0.45** | **0.94** | **0.024** |

Note. Dashes indicate reference category

Frequency of participation is: Weekly or more vs Less than weekly to participants 'How often do you [do writing, bake or cook, sew or knit, read, do hobbies, participate in a community arts group]'

Column A shows the total sample with age as a predictor; Columns B-D show age stratified analyses.

higher educational attainment and were not satisfied with aging. They were also less likely to participate in the arts if they were men and were unable to do iADLs (and difficulties with iADLs if aged 70+). All ages who had less appreciation for the arts were less likely to participate in the arts.

A sensitivity analysis stratified by two age groups to be comparable to the analyses with the other outcomes is available in S2 Table.

**Cultural event attendance.** In the sample (N = 1,465) used to examine attendance at cultural events, difficulty participating in the arts, and interested non attendees, 61.2% were women, 30.0% were educated to the college/postgraduate level, and 43.7% were retired. More than half (64.9%) reported attending a cultural event in the past 12 months, 38.4% reported difficulties participating in the arts, and 39.8% missed an event they otherwise wanted to attend.

Predictors for the whole sample are shown in Table 2. There was no evidence that cultural event attendance differed with age. When stratified by age, those aged 50–69 in the second and higher wealth quartiles ([Q2: OR = 3.26 95%CI = 1.62, 6.53], [Q3: OR = 4.18; 95% CI = 1.90, 9.21], [Q4: OR = 3.74; 95% CI = 1.57, 8.91]), and higher cognition scores (OR = 2.24; 95% CI = 1.02, 4.88) were more likely to have attended a cultural event. While individuals aged 70 and over with higher educational attainment ([high school: OR = 2.21; 95% CI = 1.23; 3.99], [college/ postgraduate: OR = 5.53; 95% CI = 2.53, 12.12]) and more frequent religious service attendance ([Monthly: OR = 2.53; 59% CI = 1.39, 4.60], [Weekly: OR = 3.12; 95% CI = 1.62, 5.99]) were more likely to have attended. Lower scores on the arts appreciation index were associated with decreased likelihood of cultural event attendance in both age groups.

## RQ2: Predictors of barriers to arts engagement

**Predictors of facing difficulty participating in the arts.** Predictors for the whole sample are shown in Table 3. Age was not associated with facing difficulty participating in the arts. When stratified by age, those aged 50–69 who frequently attended religious services ([Monthly: OR = 2.78; 95% CI = 1.59, 4.85], [Weekly: OR = 2.12; 95% CI = 1.19, 3.77]) reported difficulties engaging in the arts, while low neighbourhood safety (OR = 0.37; 95% CI = 0.18, 0.75), poor self-rated health (OR = 0.52; 95% CI = 0.29, 0.96), and less appreciation for the arts (OR = 0.53, 95% CI = 0.38, 0.75) were associated with reduced likelihood of

**Table 2. Age related differences in predictors of cultural event attendance in the past 12 months from logistic regression models.**

| | Cultural event attendance | | | | | | | | |
|---|---|---|---|---|---|---|---|---|---|
| | Total sample [A] | | | Ages 50–69 [B] | | | Ages ≥ 70 [C] | | |
| | N = 1465 | | | N = 862 | | | N = 603 | | |
| | OR | 95%CI | P | OR | 95%CI | P | OR | 95%CI | P |
| **Gender (Ref Female)** | - - | | | | | | | | |
| Male | 0.83 | 0.57 1.21 | 0.331 | 0.95 | 0.57 1.59 | 0.850 | 0.73 | 0.42 1.26 | 0.254 |
| **Race/ethnicity (Ref White)** | - - | | | | | | | | |
| Black/African American | 1.16 | 0.68 1.97 | 0.597 | 1.50 | 0.72 3.13 | 0.285 | 0.76 | 0.35 1.67 | 0.495 |
| Other ethnicity [including American Indian or Alaskan Native, Asian or Pacific Islander] | 1.15 | 0.56 2.35 | 0.702 | 0.95 | 0.41 2.17 | 0.896 | 1.50 | 0.29 7.69 | 0.630 |
| **Marital status (Ref Married)** | - - | | | | | | | | |
| Unmarried | 0.76 | 0.53 1.10 | 0.141 | 0.75 | 0.45 1.25 | 0.265 | 0.94 | 0.55 1.61 | 0.819 |
| **Educational attainment (Ref None)** | - - | | | | | | | | |
| High School/ GED | **1.76** | **1.09 2.84** | **0.020** | 1.32 | 0.60 2.91 | 0.494 | **2.21** | **1.23 3.99** | **0.008** |
| College / postgraduate | **3.38** | **1.87 6.09** | **<0.001** | 2.40 | 0.98 5.86 | 0.055 | **5.53** | **2.53 12.12** | **<0.001** |
| **Neighbourhood safety (Ref Good/excellent)** | - - | | | | | | | | |
| Fair/Poor | 1.92 | 0.93 3.97 | 0.078 | 2.38 | 0.95 5.94 | 0.065 | 1.52 | 0.46 5.03 | 0.497 |
| **Employment status (Ref Employed)** | - - | | | | | | | | |
| Unemployed/ Inactive | 0.69 | 0.40 1.19 | 0.185 | 0.78 | 0.38 1.57 | 0.476 | 0.57 | 0.21 1.53 | 0.265 |
| Retired | 0.75 | 0.48 1.16 | 0.195 | 0.68 | 0.36 1.30 | 0.244 | 0.97 | 0.44 2.14 | 0.934 |
| **Wealth, quartiled (Ref Quartile 1)** | - - | | | | | | | | |
| Quartile 2 | **1.96** | **1.13 3.39** | **0.016** | **3.26** | **1.62 6.53** | **0.001** | 0.85 | 0.36 2.00 | 0.708 |
| Quartile 3 | **2.09** | **1.20 3.64** | **0.010** | **4.18** | **1.90 9.21** | **<0.001** | 0.96 | 0.41 2.24 | 0.920 |
| Quartile 4 | **2.31** | **1.24 4.29** | **0.008** | **3.74** | **1.57 8.91** | **0.003** | 1.35 | 0.55 3.34 | 0.517 |
| **Satisfied with aging (Ref Yes)** | - - | | | | | | | | |
| No | 1.19 | 0.63 2.23 | 0.593 | 1.09 | 0.46 2.57 | 0.840 | 1.26 | 0.50 3.19 | 0.627 |
| **Satisfied with Life (Ref Yes)** | - - | | | | | | | | |
| No | 0.79 | 0.40 1.56 | 0.489 | 0.60 | 0.25 1.45 | 0.256 | 1.21 | 0.47 3.11 | 0.686 |
| **See friends (Ref Yearly/less)** | - - | | | | | | | | |
| Monthly | 1.22 | 0.61 2.45 | 0.576 | 1.31 | 0.53 3.25 | 0.551 | 1.23 | 0.49 3.10 | 0.662 |
| Weekly | 1.45 | 0.62 3.42 | 0.393 | 1.35 | 0.48 3.78 | 0.568 | 1.68 | 0.50 5.62 | 0.393 |
| **Attend religious services (Ref Yearly/Less)** | - - | | | | | | | | |
| Monthly | **1.70** | **1.11 2.62** | **0.015** | 1.36 | 0.75 2.49 | 0.312 | **2.53** | **1.39 4.60** | **0.002** |
| Weekly | **1.58** | **1.01 2.48** | **0.046** | 1.07 | 0.57 2.01 | 0.828 | **3.12** | **1.62 5.99** | **0.001** |
| **Depression CES-D (Ref None)** | - - | | | | | | | | |
| Present | 0.82 | 0.46 1.48 | 0.513 | 1.00 | 0.47 2.14 | 1.000 | 0.54 | 0.22 1.31 | 0.173 |
| **Smoker** | - - | | | | | | | | |
| Yes | 0.82 | 0.48 1.40 | 0.472 | 0.57 | 0.30 1.09 | 0.088 | 1.68 | 0.59 4.78 | 0.332 |
| **Self-rated health (Ref Good/excellent)** | - - | | | | | | | | |
| Fair/Poor | 1.17 | 0.73 1.86 | 0.518 | 1.28 | 0.67 2.45 | 0.464 | 1.11 | 0.58 2.13 | 0.743 |
| **iADL (Ref None)** | - - | | | | | | | | |
| Difficulties with activities | 0.70 | 0.44 1.14 | 0.150 | 0.74 | 0.37 1.48 | 0.386 | 0.78 | 0.40 1.52 | 0.471 |
| Unable to do activities | 1.08 | 0.38 3.06 | 0.892 | 3.31 | 0.94 11.66 | 0.062 | 0.45 | 0.12 1.75 | 0.249 |
| **Long term conditions (Ref None)** | - - | | | | | | | | |
| Yes | 0.82 | 0.55 1.20 | 0.299 | 1.01 | 0.59 1.72 | 0.981 | 0.66 | 0.37 1.16 | 0.147 |
| **Total cognition score, quartiled (Ref Quartile 1)** | - - | | | | | | | | |
| Quartile 2 | 1.34 | 0.87 2.06 | 0.190 | 1.74 | 0.86 3.52 | 0.125 | 1.06 | 0.61 1.86 | 0.828 |
| Quartile 3 | **2.03** | **1.19 3.47** | **0.009** | **2.24** | **1.02 4.88** | **0.044** | 1.79 | 0.85 3.78 | 0.127 |

*(Continued)*

**Table 2.** (Continued)

| | Cultural event attendance | | | | | | | | |
| | Total sample [A] | | | Ages 50–69 [B] | | | Ages ≥ 70 [C] | | |
| | N = 1465 | | | N = 862 | | | N = 603 | | |
| | *OR* | *95%CI* | *P* | *OR* | *95%CI* | *P* | *OR* | *95%CI* | *P* |
|---|---|---|---|---|---|---|---|---|---|
| Quartile 4 | **2.29** | **1.28** | **4.11** | **0.006** | 2.20 | 0.98 | 4.93 | 0.056 | 2.81 | 0.81 | 9.81 | 0.105 |
| **Arts index** | **0.36** | **0.27** | **0.48** | **<0.001** | 0.35 | 0.23 | 0.55 | <0.001 | 0.34 | 0.23 | 0.50 | <0.001 |

Note. Dashes indicate reference category.

Cultural event attendance is: yes vs no to 'In the past 12 months, did you go to a movie, art museum or gallery, crafts fair, or a live performance, such as a concert, play, or reading?'

Column A shows total sample; Columns B-C show age stratified analyses

reporting difficulties engaging in the arts. While in those aged 70 and older, those in the third quartile of cognition (OR = 0.51, 95% CI = 0.28, 0,95) reported less difficulties engaging in the arts. Those in less safe neighbourhoods were less likely to report difficulties in engaging in the arts in both age groups, as were individuals of Black/ African American ethnicity.

A sensitivity analysis was conducted by collapsing the outcome variable into 'neutral/agree vs disagree' (as opposed to neutral/disagree and agree) and is available in S3 Table. Results were largely the same with the addition that those with difficulties in iADLs were less likely to report difficulties participating in the arts.

**Predictors of being an "interested non-attendee" at cultural events.** Predictors for the whole sample are shown in Table 4. The odds of being an interested non-attendee did not differ in those aged 50–69 versus 70 and over. When stratified by age, individuals over 70 were more likely to be an interested non-attendee if they had higher educational attainment ([High School: OR = 2.32; 95% CI = 1.09, 4.93], [College/ postgraduate: OR = 3.40; 95% CI = 1.44, 8.00]), and less likely if their neighbourhoods were unsafe (OR = 0.33; 95% CI = 0.13, 0.81). Being in the highest quartile of cognition scores was the only other predictor of being an interested non-attendee in the 50–69 age group (OR = 2.35; 95% CI = 0.31, 0.60). Lower arts appreciation was consistently associated with lower likelihood of being an interested non-attendee in both age groups.

## Discussion

### Change in predictors of arts and cultural engagement across age

This study examined predictors of both arts and cultural engagement and barriers to engagement as people aged in the United States. For arts participation, only one factor, lower levels of interest in the arts, consistently predicted decreased likelihood of engagement across all three age groups. However, there was evidence that certain factors become predictors or became stronger predictors as people aged, such as gender, higher educational attainment, dissatisfaction with aging, difficulties with instrumental activities of daily living (iADLs), and cognition. Similarly, for cultural event attendance, lower interest in the arts predicted lack of engagement in both age groups, whereas higher educational attainment and more frequent religious service attendance became predictors as people aged. However, wealth became less important as a predictor of cultural event attendance with increasing age.

Contrary to our hypothesis, some factors such as mental and physical health were not associated with either arts participation or cultural event attendance. When examining those who had expressed an interest in participating in the arts but faced barriers to doing so, those

**Table 3. Age related differences in reported difficulties participating in the arts in the past 12 months from logistic regression models.**

| | Difficulties participating in the arts | | | | | | | | |
| --- | --- | --- | --- | --- | --- | --- | --- | --- | --- |
| | Total sample [A] | | | Ages 50–69 [B] | | | Ages ≥ 70 [C] | | |
| | N = 1465 | | | N = 862 | | | N = 603 | | |
| | OR | 95%CI | P | OR | 95%CI | P | OR | 95%CI | P |
| **Gender (Ref Female)** | -- | | | | | | | | |
| Male | 1.08 | 0.78 1.51 | 0.648 | 1.24 | 0.78 1.96 | 0.365 | 0.81 | 0.51 1.30 | 0.388 |
| **Race/Ethnicity (Ref White)** | -- | | | | | | | | |
| Black/African American | **0.49** | **0.32 0.77** | **0.002** | **0.53** | **0.29 0.98** | **0.043** | **0.42** | **0.22 0.81** | **0.010** |
| Other ethnicity [including American Indian or Alaskan Native, Asian or Pacific Islander] | 0.82 | 0.46 1.46 | 0.504 | 1.00 | 0.51 1.96 | 0.994 | 0.68 | 0.20 2.34 | 0.542 |
| **Marital status (Ref Married)** | -- | | | | | | | | |
| Unmarried | 1.12 | 0.81 1.53 | 0.495 | 0.98 | 0.62 1.55 | 0.916 | 1.13 | 0.72 1.76 | 0.595 |
| **Educational attainment (Ref None)** | -- | | | | | | | | |
| High School/ GED | 1.32 | 0.86 2.02 | 0.202 | 1.65 | 0.83 3.28 | 0.149 | 1.20 | 0.72 2.02 | 0.486 |
| College / postgraduate | 0.65 | 0.39 1.10 | 0.110 | 0.68 | 0.31 1.51 | 0.347 | 0.70 | 0.36 1.36 | 0.290 |
| **Neighbourhood safety (Ref Good/excellent)** | -- | | | | | | | | |
| Fair/Poor | **0.44** | **0.26 0.74** | **0.002** | **0.39** | **0.19 0.79** | **0.009** | **0.43** | **0.19 0.94** | **0.036** |
| **Employment status (Ref Employed)** | -- | | | | | | | | |
| Unemployed/ Inactive | 1.27 | 0.79 2.06 | 0.330 | 1.19 | 0.64 2.21 | 0.582 | 0.72 | 0.28 1.85 | 0.497 |
| Retired | 0.91 | 0.63 1.33 | 0.633 | 0.84 | 0.49 1.44 | 0.530 | 0.47 | 0.22 0.98 | 0.045 |
| **Wealth, quartiled (Ref Quartile 1)** | -- | | | | | | | | |
| Quartile 2 | 1.02 | 0.65 1.60 | 0.922 | 0.93 | 0.51 1.69 | 0.811 | 1.45 | 0.75 2.82 | 0.270 |
| Quartile 3 | 1.13 | 0.70 1.84 | 0.620 | 1.19 | 0.60 2.35 | 0.623 | 1.09 | 0.56 2.11 | 0.808 |
| Quartile 4 | 0.69 | 0.42 1.14 | 0.150 | 0.61 | 0.30 1.24 | 0.173 | 0.83 | 0.42 1.67 | 0.605 |
| **Satisfied with aging (Ref Yes)** | -- | | | | | | | | |
| No | 0.56 | 0.31 1.01 | 0.055 | 0.55 | 0.26 1.18 | 0.123 | 0.58 | 0.26 1.27 | 0.170 |
| **Satisfied with Life (Ref Yes)** | -- | | | | | | | | |
| No | 1.21 | 0.70 2.08 | 0.489 | 1.38 | 0.67 2.84 | 0.380 | 0.97 | 0.48 1.95 | 0.934 |
| **See friends (Ref Yearly/less)** | -- | | | | | | | | |
| Monthly | 1.00 | 0.56 1.80 | 0.989 | 0.94 | 0.43 2.04 | 0.867 | 1.06 | 0.44 2.55 | 0.891 |
| Weekly | 1.14 | 0.57 2.28 | 0.704 | 1.12 | 0.46 2.69 | 0.802 | 1.32 | 0.48 3.66 | 0.587 |
| **Attend religious services (Ref Yearly/Less)** | -- | | | | | | | | |
| Monthly | **1.96** | **1.31 2.92** | **0.001** | **2.78** | **1.59 4.85** | **<0.001** | 1.22 | 0.71 2.11 | 0.470 |
| Weekly | 1.38 | 0.93 2.05 | 0.105 | **2.12** | **1.19 3.77** | **0.011** | 0.66 | 0.39 1.10 | 0.109 |
| **Depression CES-D (Ref None)** | -- | | | | | | | | |
| Present | 1.30 | 0.81 2.08 | 0.272 | 1.83 | 0.99 3.38 | 0.053 | 0.70 | 0.34 1.46 | 0.342 |
| **Smoker** | -- | | | | | | | | |
| Yes | 1.36 | 0.83 2.24 | 0.221 | 1.73 | 0.92 3.24 | 0.088 | 1.10 | 0.48 2.52 | 0.814 |
| **Self-rated health (Ref Good/excellent)** | -- | | | | | | | | |
| Fair/Poor | 0.69 | 0.45 1.05 | 0.081 | **0.52** | **0.29 0.96** | **0.036** | 0.93 | 0.53 1.63 | 0.792 |
| **iADL (Ref None)** | -- | | | | | | | | |
| Difficulties with activities | 1.28 | 0.83 1.97 | 0.269 | 1.04 | 0.52 2.08 | 0.902 | 1.36 | 0.80 2.29 | 0.254 |
| Unable to do activities | 0.44 | 0.16 1.25 | 0.123 | 0.45 | 0.11 1.93 | 0.281 | 0.32 | 0.07 1.47 | 0.144 |
| **Long term conditions (Ref None)** | -- | | | | | | | | |
| Yes | **1.18** | **0.84 1.65** | **0.334** | 1.10 | 0.69 1.74 | 0.696 | 1.25 | 0.77 2.02 | 0.365 |
| **Total cognition score, quartiled (Ref Quartile 1)** | -- | | | | | | | | |
| Quartile 2 | 1.13 | 0.75 1.69 | 0.567 | 1.02 | 0.52 2.00 | 0.958 | 1.34 | 0.82 2.18 | 0.238 |
| Quartile 3 | **0.58** | **0.36 0.95** | **0.030** | 0.66 | 0.31 1.39 | 0.269 | **0.51** | **0.28 0.95** | **0.033** |

*(Continued)*

**Table 3.** (Continued)

| | Difficulties participating in the arts | | | | | | | | |
| | Total sample [A] | | | Ages 50–69 [B] | | | Ages ≥ 70 [C] | | |
| | N = 1465 | | | N = 862 | | | N = 603 | | |
| | OR | 95%CI | P | OR | 95%CI | P | OR | 95%CI | P |
|---|---|---|---|---|---|---|---|---|---|
| Quartile 4 | 0.66 | 0.40 | 1.08 | 0.099 | 0.81 | 0.39 | 1.70 | 0.583 | 0.48 | 0.22 | 1.03 | 0.058 |
| **Arts index** | **0.64** | **0.51** | **0.81** | **<0.001** | **0.53** | **0.38** | **0.75** | **<0.001** | 0.75 | 0.55 | 1.03 | 0.077 |

Note. Dashes indicate reference category.

Difficulties participating in the arts is: neutral/disagree vs agree to 'it is difficult for me to participate in the arts'

Column A presents the total sample; Columns B-C shows age stratified analyses

residing in areas with poor neighbourhood safety and individuals of Black/African American ethnicity, were associated with reduced likelihood of difficulties in both age groups. However, several factors predicted decreased likelihood of facing barriers to arts participation in the 50–69 age group only: lower arts appreciation, poor self-rated health, and religious service attendance. In the older age (70+) group, higher cognition predicted fewer barriers to arts participation. For cultural event attendance, adults with further educational attainment were more likely to be interested non-attendees, but only in the oldest age group (70+). Similarly, living in an unsafe neighbourhood predicted decreased likelihood of being an interested non-attendee in this age group, but not in the younger group. Having high levels of cognition predicted interested non-attendee status in the younger but not the older age group. Low arts appreciation was the only factor consistently associated with being an interested non-attendee in both age groups.

## Demographic and health predictors

There is a broad literature showing that engagement with the arts has a range of physical, cognitive, and mental health promoting effects [1]. Our study found a number of demographic and health predictors of age-related arts participation, many of which such as gender and difficulty with iADL in older ages have previously been associated with reduced likelihood of engagement [7, 13, 14]. While there was no clear gradient for engagement across most age groups by race/ethnicity, those in the other ethnicity category (including American Indian or Alaskan Native, Asian or Pacific Islander) were less likely than white participants to participate in the arts in older ages, which might represent further socioeconomic barriers to attendance (i.e. inability to travel to a venue). Similar findings have been reported elsewhere [25], and the interconnectedness of race/ethnicity with further factors such as income and educational attainment (contributing to structural racism) should be taken into account [35]. However, we did find Black/African Americans were less likely to report barriers in arts engagement than those in other race/ethnicity groups.

## Socioeconomic predictors

We also found that a number of socioeconomic predictors varied by whether they were related to arts or culture. Lower wealth was associated with arts participation in the youngest age group (50–59), while middle/average wealth was associated with increased likelihood of arts participation, but only in the 60–69 age group, while residing in unsafe neighbourhoods was associated with less difficulties participating in the arts. Conversely, higher wealth was associated with cultural attendance in the 50–69 age group. Similar general population findings have

**Table 4. Age related differences in predictors of interested non-attendees of cultural events in the past 12 months from logistic regression models.**

| | Interested non attendees | | | | | | | | |
| --- | --- | --- | --- | --- | --- | --- | --- | --- | --- |
| | Total sample [A] | | | Ages 50–69 [B] | | | Ages ≥ 70 [C] | | |
| | N = 1465 | | | N = 862 | | | N = 603 | | |
| | OR | 95%CI | P | OR | 95%CI | P | OR | 95%CI | P |
| **Gender (Ref Female)** | -- | | | | | | | | |
| Male | 1.13 | 0.81 1.58 | 0.479 | 1.27 | 0.83 1.94 | 0.271 | 0.73 | 0.42 1.29 | 0.279 |
| **Race/Ethnicity (Ref White)** | -- | | | | | | | | |
| Black/African American | 0.89 | 0.57 1.40 | 0.620 | 0.84 | 0.49 1.44 | 0.523 | 0.91 | 0.42 1.95 | 0.799 |
| Other ethnicity [including American Indian or Alaskan Native, Asian or Pacific Islander] | 0.62 | 0.34 1.11 | 0.107 | 0.56 | 0.28 1.14 | 0.108 | 0.94 | 0.33 2.69 | 0.908 |
| **Marital status (Ref Married)** | -- | | | | | | | | |
| Unmarried | 1.08 | 0.76 1.53 | 0.675 | 1.13 | 0.71 1.81 | 0.604 | 1.04 | 0.62 1.75 | 0.889 |
| **Educational attainment (Ref None)** | -- | | | | | | | | |
| High School/ GED | **2.02** | **1.23 3.33** | **0.006** | 1.63 | 0.83 3.20 | 0.156 | **2.32** | **1.09 4.93** | **0.028** |
| College / postgraduate | **2.52** | **1.42 4.47** | **0.002** | 1.90 | 0.88 4.08 | 0.100 | **3.40** | **1.44 8.00** | **0.005** |
| **Neighbourhood safety (Ref Good/excellent)** | -- | | | | | | | | |
| Fair/Poor | 0.69 | 0.38 1.26 | 0.227 | 0.88 | 0.41 1.88 | 0.733 | **0.33** | **0.13 0.81** | **0.016** |
| **Employment status (Ref Employed)** | -- | | | | | | | | |
| Unemployed/ Inactive | 1.31 | 0.79 2.18 | 0.297 | 1.17 | 0.63 2.17 | 0.611 | 1.87 | 0.61 5.74 | 0.272 |
| Retired | 0.95 | 0.65 1.39 | 0.805 | 0.95 | 0.57 1.59 | 0.843 | 1.66 | 0.65 4.22 | 0.288 |
| **Wealth, quartiled (Ref Quartile 1)** | -- | | | | | | | | |
| Quartile 2 | 0.98 | 0.61 1.58 | 0.926 | 0.78 | 0.43 1.41 | 0.413 | 1.58 | 0.69 3.60 | 0.278 |
| Quartile 3 | 0.78 | 0.46 1.32 | 0.355 | 0.67 | 0.34 1.33 | 0.253 | 1.13 | 0.49 2.61 | 0.770 |
| Quartile 4 | 0.61 | 0.35 1.07 | 0.083 | 0.50 | 0.24 1.06 | 0.069 | 0.97 | 0.42 2.24 | 0.949 |
| **Satisfied with aging (Ref Yes)** | -- | | | | | | | | |
| No | 0.96 | 0.54 1.70 | 0.887 | 1.02 | 0.51 2.06 | 0.954 | 0.78 | 0.34 1.76 | 0.540 |
| **Satisfied with Life (Ref Yes)** | -- | | | | | | | | |
| No | 0.63 | 0.37 1.08 | 0.093 | 0.63 | 0.32 1.24 | 0.178 | 0.69 | 0.32 1.49 | 0.341 |
| **See friends (Ref Yearly/less)** | -- | | | | | | | | |
| Monthly | 1.46 | 0.77 2.78 | 0.245 | 1.55 | 0.72 3.33 | 0.266 | 1.30 | 0.43 3.92 | 0.641 |
| Weekly | 1.48 | 0.72 3.02 | 0.282 | 1.57 | 0.68 3.61 | 0.291 | 1.36 | 0.40 4.65 | 0.619 |
| **Attend religious services (Ref Yearly/Less)** | -- | | | | | | | | |
| Monthly | 0.93 | 0.62 1.41 | 0.737 | 0.88 | 0.53 1.47 | 0.620 | 0.93 | 0.47 1.83 | 0.836 |
| Weekly | 0.95 | 0.63 1.44 | 0.818 | 0.80 | 0.46 1.38 | 0.414 | 1.14 | 0.61 2.12 | 0.679 |
| **Depression CES-D (Ref None)** | -- | | | | | | | | |
| Present | 1.27 | 0.74 2.18 | 0.381 | 1.49 | 0.75 2.95 | 0.253 | 0.86 | 0.38 1.95 | 0.714 |
| **Smoker** | -- | | | | | | | | |
| Yes | 1.19 | 0.72 1.98 | 0.492 | 0.97 | 0.53 1.79 | 0.933 | 1.80 | 0.63 5.15 | 0.271 |
| **Self-rated health (Ref Good/excellent)** | -- | | | | | | | | |
| Fair/Poor | 0.89 | 0.56 1.41 | 0.619 | 0.84 | 0.46 1.55 | 0.584 | 1.24 | 0.65 2.37 | 0.519 |
| **iADL (Ref None)** | -- | | | | | | | | |
| Difficulties with activities | 1.35 | 0.86 2.13 | 0.195 | 1.72 | 0.88 3.38 | 0.115 | 1.12 | 0.62 2.02 | 0.698 |
| Unable to do activities | 1.52 | 0.51 4.50 | 0.452 | 1.55 | 0.34 7.03 | 0.573 | 1.52 | 0.32 7.24 | 0.597 |
| **Long term conditions (Ref None)** | -- | | | | | | | | |
| Yes | 0.99 | 0.69 1.40 | 0.938 | 0.87 | 0.56 1.35 | 0.536 | 1.65 | 0.92 2.96 | 0.093 |
| **Total cognition score, quartiled (Ref Quartile 1)** | -- | | | | | | | | |
| Quartile 2 | 1.34 | 0.85 2.12 | 0.202 | 1.47 | 0.73 2.96 | 0.276 | 1.10 | 0.62 1.98 | 0.743 |
| Quartile 3 | 1.20 | 0.70 2.05 | 0.501 | 1.37 | 0.64 2.91 | 0.418 | 0.79 | 0.39 1.58 | 0.503 |

*(Continued)*

**Table 4.** (Continued)

|  | Interested non attendees | | | | | | | | |
|---|---|---|---|---|---|---|---|---|---|
|  | Total sample [A] | | | Ages 50–69 [B] | | | Ages ≥ 70 [C] | | |
|  | N = 1465 | | | N = 862 | | | N = 603 | | |
|  | OR | 95%CI | P | OR | 95%CI | P | OR | 95%CI | P |
| Quartile 4 | 2.03 | 1.20  3.42 | 0.008 | 2.35 | 1.10  5.02 | 0.027 | 1.17 | 0.56  2.45 | 0.678 |
| **Arts index** | **0.38** | **0.29  0.48** | **<0.001** | **0.43** | **0.31  0.60** | **<0.001** | **0.27** | **0.18  0.39** | **<0.001** |

Note. Dashes indicate reference category.

Interested non-attendees is yes vs no to 'In the last 12 months, was there an event of this type that you wanted to go to but did not'

Column A presents total sample; Columns B-C shows age stratified analyses

also identified these socioeconomic differences [15], which may suggest socioeconomic inequalities in older ages are greater for cultural attendance relative to arts engagement due to additional costs (e.g. tickets) and transportation required, while arts activities can more easily be undertaken at home and therefore potentially be more affordable. Future studies may wish to examine home-based cultural attendance activities (e.g., virtual museum tours) to identify if these differences are still observed; this may be of particular interest in light of the COVID-19 pandemic during which a large number of cultural activities have become digitalised. However, it is possible that digitalisation of these events could bring further age related inequalities among those with low digital literacy [36]. Additionally, education became an increasingly important predictor of participation and event attendance as people aged. This could also be an indication of cognitive capacity as adults with higher educational attainment are more likely to have better preserved cognition, which could enhance continued arts and cultural participation, which in turn is associated with better cognition [7, 37]. However, our measure of cognition was not consistently associated with participation or engagement, suggesting this does not fully explain the finding. So, this remains to be explored further.

## Psychosocial predictors

We also examined a number of psychosocial predictors of engagement. Dissatisfaction with aging was a clear predictor for engagement with the arts for adults over the age of 60, which may support previous research showing that older adults (particularly following retirement) turn to the arts to aid with boredom, loneliness, and social connectedness [38, 39]. Arts and cultural appreciation are known to enhance social capital [40], however we found low levels of arts appreciation was a relatively consistent predictor of arts and cultural engagement across ages, suggesting it is an embedded predictor. Further, low appreciation for the arts was a constant predictor of not being an interested non-attendee and non-participant, reinforcing this concept. There was a difference observed in religious service attendance, with adults in the 70 + age group (but not the ages 50–69 group) frequently attending services reporting more frequent cultural event attendance. It is possible that the frequency of religious attendance increases social capital which in turn may encourage more community activity in a virtuous cycle [41]. Furthermore, some specific activities, such as live performances of singing, occur during some religious services and are often held in places of worship. Therefore, it is also possible that some older adults better retain the mobility and capacity to engage in community activities such as religious service attendance and cultural events, such that these activities correlate increasingly as people age. Unexpectedly, there were no patterns of association observed for depression status with engagement or barriers to engagement in arts and cultural events.

## Implications

While there are already a number of known barriers to arts and cultural engagement, it is important to identify how these inequalities may continue to widen with age. Some factors identified may be multifaceted, such as race/ethnicity, which is interconnected with numerous socioeconomic factors (i.e. income and educational attainment) [35]. Additionally, there were barriers identified for cultural engagement which were not observed for the arts suggesting this may arise from additional social, financial, and transportation linked to attendance (as cultural engagement tends to be passive, i.e. museums, theatre etc.). Therefore, it is important to find ways to make cultural activities more accessible to people of all ages. For example, virtual theatres during the COVID-19 pandemic were able to reach wider audiences [42, 43]. Promoting these opportunities to older adults may help increase awareness and uptake if they continue to be digitalised.

Adults who are isolated and lonely may avoid social situations for fear of vulnerability and further rejection [44]. Community outreach groups that are inclusive of all ages may be helpful to connect younger and older adults to attend these activities together [45–47]. Such community groups may also help overcome barriers such as transportation and costs (i.e., reduced group costing). Transportation barriers for older adults may be especially problematic in the US, primarily resulting from issues in built and social environment, transportation services, and individual attributes (e.g., a person's ability to walk or rely on friends) [48], and research has repeatedly noted the need for accessible and affordable option for older adults [48–50]. Conversely, we found that there were fewer barriers for engagement with the arts, suggesting these activities are more likely once people can engage with at home, when alone, or that they may already be embedded within their regular cultural routines (i.e., singing for religious practices). Active arts activities, such as crafting or sewing, have been associated with longevity in older adults [13]. Increasing awareness of the range of health benefits associated with arts and cultural engagement is therefore important [1]. This can be done in the primary care settings, where for example, in the United Kingdom, social prescribing structures facilitate connecting people to community groups for arts activities, cookery, and gardening) [51].

## Strengths and limitations

There are a number of strengths to the current study. HRS is large nationally representative cohort study of retirement and old age in the United States. The rich data collected in HRS on different aspects of aging (i.e., mental, physical, and cognitive) alongside the range of sociodemographic data available has allowed us to examine a number of potential predictors of age-related engagement with arts and cultural attendance. HRS has updated their variable choice and inclusive language across demographics since study inception. For example, increasing the number of race/ethnicity responses and allowing participants to choose more than one option, subsequently providing accurate choices to increase visibility of survey respondents and prevent social exclusionism through overgeneralisation into broad unrepresentative categories. Additionally, the 2014 Culture and Arts module provided a range of outcomes to examine, including both frequency of and difficulties in arts and cultural engagement. However, there are also a number of limitations to consider. The sample of individuals responding to the 2014 Culture and Arts module was small with respect to the total cohort sample, and therefore we were unable to stratify by more than two age groups. In doing so, we may have lost some variation by predictors between smaller age groupings. Similarly, we had to collapse some outcomes and predictors due to the small samples, for example we included 'neutral' with disagree when examining whether individuals reported difficulties engaging in the arts. However, we ran a sensitivity analysis including neutral along with agree and found results to be largely

consistent with the current results with the additional inclusion of iADLs. Additionally, some survey questions were quite broadly phrased and are therefore difficult to make strong inferences from the data based on potential variation in interpretation across participations. For example, neighbourhood safety is based on a single self- reported question but we are unable to determine why participants may have felt unsafe (i.e. high levels of crime or lack of safe transportation options). Finally, these analyses were cross-sectional and therefore we cannot make causal inferences from the findings.

## Conclusions

Our results suggest that socioeconomic inequalities in arts and cultural engagement may increase in older ages, with arts activities overall remaining more accessible (compared to cultural engagement). Given the numerous associations of arts and cultural engagement with mental and physical health outcomes in older age [11, 13, 37], reducing these inequalities is important to supporting health equity amongst older adults. The largest barriers to engagement we identified were for cultural engagement and likely involved transportation requirements. Therefore, this issue, along with upstream issues of structural racism and interconnected systems that embed inequities in systems and policies, must be explored further. Subsidized ticket prices for older adults and/or identification of how access to these events can be more home-based and offering accessible transportation are possibilities. Further, opportunities to increase awareness in primary care, such as social prescribing in the United Kingdom, has been proven beneficial in older adults [51]. Similar programs and policy, if adopted in the US, may aid in reducing age-related inequalities in access and ensure individuals of all ages are able to benefit from arts and cultural engagement and are encouraged as the focus of future research.

## Supporting information

**S1 Table. Missing data across arts outcomes.**
(DOCX)

**S2 Table. Age related differences in predictors of frequency of arts participation- two age categories from logistic regression models.**
(DOCX)

**S3 Table. Age related differences in reported difficulties participating in the arts- neutral/ agree vs disagree from logistic regression models.**
(DOCX)

## Acknowledgments

We thank Shanae Burch, thought leader on work at the intersections of the arts, equity, and public health in the US, for her comments on this manuscript. We also gratefully acknowledge the contribution of the HRS study participants.

## Author Contributions

**Conceptualization:** Meg Fluharty, Elise Paul, Jessica Bone, Feifei Bu, Daisy Fancourt.

**Formal analysis:** Meg Fluharty.

**Funding acquisition:** Jill Sonke, Daisy Fancourt.

**Investigation:** Meg Fluharty.

**Methodology:** Meg Fluharty.

**Supervision:** Elise Paul, Daisy Fancourt.

**Writing – original draft:** Meg Fluharty.

**Writing – review & editing:** Elise Paul, Jessica Bone, Feifei Bu, Jill Sonke, Daisy Fancourt.

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
