## [Decision Letter · Decision Letter 0]

10 Aug 2021

PONE-D-21-14706

Difference in predictors and barriers to arts and cultural engagement with age in the United States: A cross-sectional analysis using the Health and Retirement Study

PLOS ONE

Dear Authors,

Thank you for submitting your manuscript to PLOS ONE. After careful consideration, we feel that it has merit but does not fully meet PLOS ONE’s publication criteria as it currently stands. Therefore, we invite you to submit a revised version of the manuscript that addresses the points raised during the review process.

We look forward to receiving your revised manuscript.

Kind regards,

Marcel Pikhart

Academic Editor

PLOS ONE

Journal Requirements:

Reviewers' comments:

Reviewer's Responses to Questions

**Comments to the Author**

1. Is the manuscript technically sound, and do the data support the conclusions?

Reviewer #1: Yes

Reviewer #2: Yes

2. Has the statistical analysis been performed appropriately and rigorously? 

Reviewer #1: Yes

Reviewer #2: Yes

3. Have the authors made all data underlying the findings in their manuscript fully available?

Reviewer #1: Yes

Reviewer #2: Yes

4. Is the manuscript presented in an intelligible fashion and written in standard English?

Reviewer #1: Yes

Reviewer #2: Yes

5. Review Comments to the Author

Reviewer #1: This manuscript deals with a topic that is quite relevant in the current context.

The introduction is well written, and I recommend that the authors insert the study's hypotheses at the end of the introduction. I also suggest that they provide a more robust justification for doing this work.

Regarding the method, I recommend that the authors insert the inclusion and exclusion criteria of the participants and the instruments used to identify the constructs.

The data analysis should also be better described, presenting all the statistical procedures used in work. I recommend that the authors make a detailed description of the tables throughout the text.

The Discussion needs to be deepened, comparing the results found in this study with others in the literature.

Reviewer #2: In this article, the authors provided interesting insights to predictors and barriers to participation and engagement with art and how these may change with aging. As a researcher who focuses on the basic research of aging, it was great to read and learn about more wholistic contributors to the aging process and about potential impediments to aging individuals to take advantage of available option that can affect healthy aging. The article is well-written and thought provoking. However, there are a few minor adjustments that the authors should consider prior to publishing the article.

1. As PLOS ONE readers come from diverse backgrounds and areas of expertise, the authors should justify and explain the reasons for collapsing the responses which results in binary data (under methods -> Outcome variables)

2. The first discussion paragraph is simply a restatement of the results and does not add to the discussion. That is also somewhat true for the second discussion paragraph as well. In fact, the first sentence that provides critical assessment of the data comes only in the 7th sentence of the 3rd discussion paragraph. The authors should consider rewriting some of the discussion section to reflect a stronger critical assessment of the data and its relevancy to everyday life and health outcomes of aging individuals.

Other than these two comments, I think that this short manuscript is a great addition to PLOS ONE and that many readers may find it useful.

6. PLOS authors have the option to publish the peer review history of their article (what does this mean?). If published, this will include your full peer review and any attached files.

Reviewer #1: No

Reviewer #2: **Yes: **Arik Davidyan Ph.D.

---

## [Author Response · Author response to Decision Letter 0]

17 Sep 2021

We would like to thank the editors and reviewers for the opportunity to revise and resubmit, and for the helpful comments which have strengthen our manuscript. We have responded to each comment point-by-point below. 

Reviewer #1: This manuscript deals with a topic that is quite relevant in the current context.

The introduction is well written, and I recommend that the authors insert the study's hypotheses at the end of the introduction. I also suggest that they provide a more robust justification for doing this work.

Many thanks for the kind comments and are pleased you found this topic relevant and the writing clear. 

We have now added a sentence to more clearly introduce our study justifications, which are further outlined in paragraphs two and three of the introduction section. These justifications centre around the observation that while we know there are predictors of and barriers to arts and cultural engagement- there is little research on how these factors change as people age. Further, there has been little research into engagement versus wanting to engage (but being unable to do so) which this paper also examines. Further, we add to existing literature by examining predictors and barriers for different age groups of older adults, as well as active rather than passive arts engagement. 

We have also expanded the hypotheses to make this clearer. 

Regarding the method, I recommend that the authors insert the inclusion and exclusion criteria of the participants and the instruments used to identify the constructs.

We have now included a statement in the methods to make clearer our process for selecting sample participants. This statement clearly states the inclusion criteria as well as a numeric breakdown of sample selection. If any validated instruments were used in the study (for example the CES- D or iADL), they were appropriately referenced. Therefore, the remaining variables were taken from questions which were available to us in the HRS survey and are summarised in the methods section. 

The data analysis should also be better described, presenting all the statistical procedures used in work. I recommend that the authors make a detailed description of the tables throughout the text.

We have now updated the statistical analysis section and added text under each table to further

clarify the statistical procedures used. 

The Discussion needs to be deepened, comparing the results found in this study with others in the literature.

We have now expanded upon the discussion with additional references and comparisons to previous literature, relevance of the findings, and suggested policy implications. 

Reviewer #2: In this article, the authors provided interesting insights to predictors and barriers to participation and engagement with art and how these may change with aging. As a researcher who focuses on the basic research of aging, it was great to read and learn about more wholistic contributors to the aging process and about potential impediments to aging individuals to take advantage of available option that can affect healthy aging. The article is well-written and thought provoking. However, there are a few minor adjustments that the authors should consider prior to publishing the article.

1. As PLOS ONE readers come from diverse backgrounds and areas of expertise, the authors should justify and explain the reasons for collapsing the responses which results in binary data (under methods -> Outcome variables)

We have now noted in our description of the outcome variables of the methods section that small sample size was the reason for our collapsing into binary categories. 

2. The first discussion paragraph is simply a restatement of the results and does not add to the discussion. That is also somewhat true for the second discussion paragraph as well. In fact, the first sentence that provides critical assessment of the data comes only in the 7th sentence of the 3rd discussion paragraph. The authors should consider rewriting some of the discussion section to reflect a stronger critical assessment of the data and its relevancy to everyday life and health outcomes of aging individuals.

We agree with the reviewers that a large portion of the discussion is focused on restating the results. However, as this study examined a large number of predictors and barriers to engagement across several outcomes, it is important these are summarised for our readers. To aid readability, we have now disaggregated these summaries and added subheadings. We have also extended the comparisons to previous literature and added text on the relevancy to life and health of ageing people. 

Other than these two comments, I think that this short manuscript is a great addition to PLOS ONE and that many readers may find it useful.

Thank you for your kind comments about the manuscript, and we believe the new additions have further strengthened it.

---

## [Decision Letter · Decision Letter 1]

6 Dec 2021

Difference in predictors and barriers to arts and cultural engagement with age in the United States: A cross-sectional analysis using the Health and Retirement Study

PONE-D-21-14706R1

Dear Authors,

We’re pleased to inform you that your manuscript has been judged scientifically suitable for publication and will be formally accepted for publication once it meets all outstanding technical requirements.

Kind regards,

Marcel Pikhart

Academic Editor

PLOS ONE

Additional Editor Comments (optional):

Reviewers' comments:

Reviewer's Responses to Questions

**Comments to the Author**

1. If the authors have adequately addressed your comments raised in a previous round of review and you feel that this manuscript is now acceptable for publication, you may indicate that here to bypass the “Comments to the Author” section, enter your conflict of interest statement in the “Confidential to Editor” section, and submit your "Accept" recommendation.

Reviewer #1: All comments have been addressed

2. Is the manuscript technically sound, and do the data support the conclusions?

Reviewer #1: Yes

3. Has the statistical analysis been performed appropriately and rigorously? 

Reviewer #1: Yes

4. Have the authors made all data underlying the findings in their manuscript fully available?

Reviewer #1: Yes

5. Is the manuscript presented in an intelligible fashion and written in standard English?

Reviewer #1: Yes

6. Review Comments to the Author

Reviewer #1: (No Response)

7. PLOS authors have the option to publish the peer review history of their article (what does this mean?). If published, this will include your full peer review and any attached files.

Reviewer #1: No

---

## [Editor Report · Acceptance letter]

9 Dec 2021

PONE-D-21-14706R1 

Difference in predictors and barriers to arts and cultural engagement with age in the United States: A cross-sectional analysis using the Health and Retirement Study 

Dear Dr. Fluharty:

I'm pleased to inform you that your manuscript has been deemed suitable for publication in PLOS ONE. Congratulations! Your manuscript is now with our production department. 

Kind regards, 

on behalf of

Dr. Marcel Pikhart 

Academic Editor

PLOS ONE